# Sources and Determinants of Discretionary Food Intake in a Cohort of Australian Children Aged 12–14 Months

**DOI:** 10.3390/ijerph17010080

**Published:** 2019-12-20

**Authors:** Claire Coxon, Gemma Devenish, Diep Ha, Loc Do, Jane A. Scott

**Affiliations:** 1School of Public Health, Curtin University, Perth, WA 6102, Australia; 2Australian Research Centre for Population Oral Health, The University of Adelaide, Adelaide, SA 5000, Australia

**Keywords:** discretionary foods, determinants, food sources, toddlers

## Abstract

Despite recommendations to the contrary, consumption of discretionary (energy-dense, nutrient-poor) foods begins for some children early in the weaning period, and the proportion of children consuming discretionary foods increases markedly in the second year of life. The purpose of this study was to determine intake and sources of discretionary foods in a cohort of 828 Australian toddlers (mean age: 13.1mo), and to identify determinants of discretionary food intake. At approximately 12 months of age, 3 non-consecutive days of dietary intake data were collected using a 24-h recall and 2-day food record, and the percentage total energy derived from discretionary foods was estimated. Linear regression was used to identify associations between discretionary food intake and socio-demographic determinants (mother’s age, level of education, country of birth, pre-pregnancy body mass index, socioeconomic position, parity, age of child when mother returned to work, and child’s sex) and age at which complementary foods were introduced. The average energy intake of children in this cohort was 4040 (±954.7 SD) kJ with discretionary foods contributing an average of 11.2% of total energy. Sweet biscuits, and cakes, muffins, scones and cake-type desserts contributed 10.8% and 10.2% of energy intake from discretionary foods, respectively. Other key contributors to energy intake from discretionary foods included sausages, frankfurters and saveloys (8.3%), vegetable products and dishes where frying was the main cooking technique (8.6%), butter (7.3%), and finally manufactured infant sweet or savory snack foods (9.3%). Higher intakes of discretionary food were associated with children having two or more siblings (*p* = 0.002), and being born to younger mothers (<25 years) (*p* = 0.008) and mothers born in Australia or the United Kingdom (*p* < 0.001). Parents, in particular young mothers and those with larger families, need practical guidance on how much of, and how often, these foods should be eaten by their children.

## 1. Introduction

Early life is characterized by rapid growth and development, both of which are influenced by the high susceptibility of body systems and tissues to external factors such as poor nutrition [1]. Dietary intake in the first two years of life is critical in establishing life-long food preferences and foundations for health in later childhood and adulthood [2,3,4,5]. It is well established that over-nutrition is a major public health problem in high income nations [6], with levels of overweight and obesity in preschoolers almost doubling globally since the 1990s [7]. The 2017/18 National Health Survey identified that just under one quarter of Australian children aged 2 to 4 years were overweight or obese [8]. 

The Australian Infant Feeding Guidelines (IFG) recommend complementary foods be introduced at around 6 months of age while continuing breastmilk or formula, increasing in quantity and variety so that by 12 months children are consuming family foods [9]. It is recommended that parents provide a range of foods from the five core food groups to support optimal growth and development. Energy dense, nutrient poor foods, referred to in the Australian Dietary Guidelines (ADGs) as discretionary foods, should be avoided during this transition period. Discretionary foods, as defined by the ADGs, are those high in added sugars, added salt, saturated fat and alcohol, and include cakes, biscuits, confectionary, pastries, ice-cream, butter, cream, spreads high in saturated fat and/or added sugar, potato chips and other salty snack foods and sugar-sweetened beverages including cordials, soft-drinks, sports drinks and energy drinks [10]. These foods are highly palatable and although they provide a sense of enjoyment and variety [10], discretionary foods are not nutritionally necessary and increase the risk of poor health outcomes for children such as dental caries, weight gain and cardiovascular risk factors [11,12,13,14,15,16,17].

As Australian national dietary surveys have focused only on the intake of children 2 years and over, data on the dietary intake of Australian children younger than 2 years are limited. A small number of studies have assessed the discretionary food intake, defined in earlier studies as ‘extras’ or non-core foods, of Australian infants and toddlers younger than 2 years and reported that the majority of children were consuming discretionary foods well before 2 years of age [18,19,20,21], with sweet biscuits (cookies), hot chips (French fries), ice-cream and sweet and savory snack foods being the most commonly consumed items [18,19,22,23]. Furthermore, as children become older, discretionary foods contribute an increasing proportion of the total energy intake [21,24,25]. The 2011-2012 Australian Health Survey (AHS) found that for children 2–3 years of age, approximately 30% of total energy intake came from discretionary foods [26], with sweet biscuits, sugar-sweetened beverages and savory snack foods being consumed on the day of the survey by 37.5%, 29.5% and 17.0% of children in this age group, respectively [27]. 

As the relationship between discretionary food intake and chronic disease is well established [28], and the first two years of life is recognized as a nutritionally vulnerable period, it is necessary to gain further insight into complementary feeding behaviors associated with discretionary food intake among Australian children under 2 years of age. Therefore, the current study aimed to (1) estimate the contribution discretionary foods make to the total energy intake of the diets of a cohort of Australian children aged 12 to 14 months; (2) identify the types of discretionary foods consumed by this cohort; and (3) identify the sociodemographic determinants associated with consumption of discretionary foods. This is the first study of Australian children under the age of 2 years of age to use the methodology employed in the 2011–2012 AHS to flag discretionary foods [29], allowing direct comparison with the intakes of older Australian children [27].

## 2. Materials and Methods 

### 2.1. Participant Recruitment and Data Source

This study is an analysis of dietary data collected as part of the longitudinal Study of Mothers’ and Infants’ Life Events affecting oral health (SMILE) [30]. This birth cohort study is following a group of socioeconomically-diverse mother-infant dyads with the aim to identify sociodemographic factors and early life events associated with early childhood caries and obesity. Originally designed to follow children from birth into their third year of life, SMILE has received additional funding to follow the cohort until 7 years of age. 

Recruitment took place between July 2013 and August 2014 from three major hospitals with maternity wards in Adelaide, South Australia, with 2147 mothers and 2181 children including 34 sets of twins being recruited. Mothers with sufficient English competency and those not intending to move outside of the greater Adelaide area in the next 12 months were invited to participate in the study. Recruitment targets were set for each hospital based on the proportion of infants delivered at each hospital annually. Women delivering in the hospital servicing the most socially disadvantaged areas of Adelaide were oversampled by roughly 50% to offset the expected higher attrition rate for these areas [30]. 

Socio-demographic data and data on current feeding practices were collected at recruitment, and data on diet and feeding practices, oral health behaviors and practices and dental visiting patterns was collected when the child was 3, 6, 12 and 24 months of age. All questionnaires were self-completed by mothers as either online or hard copy questionnaires emailed or mailed to mothers, respectively. Children underwent a dental examination under standardized conditions between their second and third birthdays [30].

### 2.2. Collection and Handling of Dietary Data at 12 Months of Age

For logistical reasons and to reduce both participant and investigator burden, dietary data used in this study were collected via a 24-hour dietary recall (24 h-DR) and a two-day food record. Once children turned one year of age a food record booklet containing detailed instructions for its completion and information about the upcoming 24 h-DR were mailed to the 1921 mothers remaining in the study. The 24 h-DR was completed via telephone by one of two trained dietitians using the five-step multiple pass method [31] with reference made to the portion size images and household measures included in the food record booklet to assist in estimation of dietary intake. These images had been used previously in national dietary surveys of children [31,32] and included actual size photographs of graduated household measuring cups and spoons, with additional images of spoons filled to varying levels and bowls with lines indicating fill-levels. The detailed information collected by interviewers in the 24 h-DR provided mothers with an example of the level of detail required when they completed the two-day food record. Upon completion of the interview, mothers were assigned two non-consecutive days, one weekday and one weekend day, within the next 10-day period, to complete a food record of their child’s dietary intake. Those with three complete days of reported dietary intake data were included in the current study (*n* = 828). 

Dietary data were entered into FoodWorks version 8 (Xyris Software, Australia Pty Ltd, Brisbane, Australia) for analysis using the AUSNUT 2011-13 food composition database [33]. Data were double-entered by trained nutritionists/dietitians using protocols and calibration procedures for data entry to ensure standardization. When detail was lacking in the food record, the 24 h-DR was used for clarification purposes. Due to the AUSNUT 2011-13 database containing a limited number of commercial infant foods, a total of 187 manufactured infant and toddler foods reported to have been consumed by children in the study were added to the database according to the nutrient information available on the product’s nutritional information panel or the manufacturer’s website. Some examples of such foods include infant and toddler snack bars, yoghurts, custards, fruit gels, and cereal-based savory and sweet snacks. New foods were assigned an 8-digit food code according to the AUSNUT 2011-13 naming conventions. Foods were classified as discretionary or non-discretionary according to the ABS discretionary flag list at the individual food item level (8-digit code) [29] (Box 1), as specified in the ADGs [10] and supporting documents [34]. The 187 infant foods not listed in the AUSNUT 2011-13 food composition database or the Australian Bureau of Statistics (ABS) discretionary flag list, were individually assessed and categorized as discretionary or non-discretionary using the same criteria as outlined in the AHS User Guide [29] (Appendix A). 

Box 1Principles for identifying discretionary foods employed by the Australian Bureau of Statistics. Source: Australian Bureau of Statistics, Australian Health Survey User Guide [29].The main principle used to classify foods as discretionary is that they were specified or inferred in the 2013 Australian Dietary Guidelines [10] and supporting documents [34] as discretionary. These included cakes, biscuits pastries, ice-cream, butter, cream, spreads high in saturated fat and/or added sugar, potato chips and other salty snack foods and sugar-sweetened beverages including cordials, soft-drinks, sports drinks and energy drinksIn addition
fruit juices (100%) are classified as non-discretionary; other juice drinks are classified as discretionaryall soft drinks are classified as discretionary, including intense sweetened drinksall confectionery is classified as discretionaryThe following additional criteria based on nutrient profiles were used to help identify foods as non-discretionary or discretionary at the food code level. These criteria are based on cut-offs used in the modelling that supported the development of the ADGs:
for breakfast cereals, discretionary foods are defined to be those breakfast cereals with >30 g sugar per 100 g or for breakfast cereals with added fruit >35 g sugar/100 gfor mixed dishes with cereal content (e.g. sandwiches, burgers, wraps, sushi, pizzas) discretionary foods are defined to be those with >5 g sat fat per 100 g. Use of the 5 g saturated fat/100 g cut-off is consistent with the ADGs recommendation.all milk based drinks are defined to be non-discretionary, including flavored milks and those made up from dry powders such as hot chocolate powdertea and coffee beverage products sold with added sugar are flagged as discretionaryall soup dry mixes are flagged as discretionary due to their high sodium content/100 g, noting the dry mix can be used in other dishes. Dry soup mix made up with water is non-discretionary as it has similar sodium content to other ready to eat soups.

### 2.3. Statistical Analyses

Data from FoodWorks were exported to Microsoft Access (Microsoft Office 2016, Albuquerque, NM, USA) and energy (kJ), total fat (g), saturated fat (g), free sugars (g), and sodium (mg) from all foods and discretionary foods only were summed and averaged for the three days of dietary data provided for each of the 828 participants. Dietary data were then imported into SPSS Version 25.0 (IBM Corp, Armonk, NY, USA) and merged with the sociodemographic data collected at recruitment for statistical analysis. The percent total energy (%TotE), total fat, saturated fat, free sugars and sodium coming from discretionary foods were derived from the averaged values. Data are reported descriptively and include means, standard deviations (SD), ranges, 25th percentiles and 75th percentiles. 

The percentage contribution to %TotE from discretionary foods, grouped according to food groups, was calculated for all participants, and the percentage of participants consuming each of the discretionary food groups was also determined. This was completed at the sub-major (3-digit) food group levels for the majority of foods, with the minor food group (5-digit) levels also being used for manufactured infant foods only, to identify the types of infant food products contributing to discretionary food intake among Australian toddlers.

Explanatory variables assessed as possible predictors of %TotE from discretionary food included maternal age (<25, 25–34, and ≥35 years); maternal education (high school/vocational or some university and above); maternal country of birth (Australia and New Zealand, United Kingdom, India, China, Asia-Other, and Other); maternal pre-pregnancy BMI (<25, 25–24.99, and ≥30 kg/m^2^); number of children (1, 2, and ≥3 children); child sex (male and female) and age of introduction of complementary foods (<17, 17–25, and ≥26 weeks). Residential postcodes were used to assign a measure of socio-economic position (SEP) using the Index of Relative Socio-Economic Advantage and Disadvantage (IRSAD) deciles which were collapsed into 5 groups (deciles 1–2, deciles 3–4, deciles 5–6, deciles 7–8, and deciles 9–10) [35].

The General Linear Model (GLM) procedure in SPSS was used to run multiple linear regression analysis to investigate the association between explanatory factors and %TotE from discretionary foods. Factors that were associated with %TotE from discretionary foods (*p* ≤ 0.10) in the simple linear regression analyses were simultaneously entered into the multiple linear regression model to identify independent associations. Cook’s distance values that were larger than 0.5 and Studentized deleted residuals larger than 4.0 were considered to be outliers [36]. P values of less than 0.05 were considered to indicate statistical significance.

In order to account for extreme under- and over-reporting of dietary intake, sensitivity analyses were conducted with participants identified as having plausible energy intake. Plausible status was determined using the ratio of reported energy intake (EI) and an age and gender specific estimated energy requirement (EER). Participants were said to have plausible intakes if they had a ratio of EI:EER between 0.54 and 1.46 [37].

There were minimal missing data for sociodemographic variables, and no attempt was made to impute missing data. A previous sensitivity analysis for other dietary outcomes investigated in this cohort, in which missing data for sociodemographic explanatory variables was imputed under the assumption that data were missing at random, revealed that distributions of variables in the imputed data sets were consistent with the complete case data [38]. 

### 2.4. Ethical Considerations

The study was approved by the Southern Adelaide Clinical Human Research Ethics Committee (HREC/50.13, approval date: 28 February 2013) and the South Australian Women and Children Health Network (HREC/13/WCHN/69, approval date: 7 August 2013). Signed informed consent was obtained from all mothers.

## 3. Results

### 3.1. Participant Characteristics

The mean age of children was 13.1 months (±1.0 SD). Maternal and child characteristics are presented in Table 1. The majority of mothers were between the ages of 25 and 34 (mean 30.8 ± 4.9 SD) years, born in Australia and New Zealand, had received some university level education or above, were primiparous and were in the healthy weight range. Socio-economic position measured by IRSAD was relatively evenly spread over the five categories ranging from the least disadvantaged (14.5%) to the most socially advantaged (23.6%) (Table 1). 

Compared to participants in this analysis, mothers who provided baseline data but did not provide complete dietary data at 12 months were younger (*p* < 0.001), less educated (*p* < 0.001), more socially disadvantaged (*p* < 0.001) and more likely to have been born outside of Australia (*p* < 0.001) [37]. However, due to the deliberate over-recruitment of disadvantaged women into the SMILE study to compensate for anticipated higher attrition among this group [30] the analysis population was generally representative of the South Australian maternity population for 2013 [39].

### 3.2. Discretionary Food Intake and Contribution to Total Energy and Selected Nutrient Intakes

Of the 828 children with complete dietary data, 793 (95.7%) consumed discretionary foods on one or more of the three monitored days, leaving only 35 (4.3%) children who did not consume discretionary foods on any of the three days. A total of 703 children had plausible energy intakes, of which 669 (95.2%) consumed discretionary food on one or more of the three days.

The average energy intake of children in this cohort was 4040 (±954.7 SD) kJ with discretionary foods contributing an average of 11.2% total energy (Table 2). Percent total fat (14.0%) and saturated fat (14.2%) from discretionary foods were similar, whilst 22.6% of sodium and 46.1% of free sugars intake came from discretionary foods.

### 3.3. Food Group Contribution to Discretionary Food intake

The per capita contribution of discretionary foods, by food subgroup, to total energy, and the percentage of children consuming each food subgroup are reported in Table 3. 

The most commonly consumed discretionary foods were cereal based products and dishes, with 61.5% of children consuming one or more foods from this group and 36.6% of children consuming sweet biscuits specifically. Fats and oils were the next most commonly consumed food group with 41.5% of children consuming butter whilst other commonly consumed groups included processed meats (28.3%), gravies and savory sauces (28.3%), fried potatoes (19.1%), and miscellaneous products, where yeast-based spreads were the most commonly consumed (36.0%). When considering manufactured infant foods, 33.2% of children consumed discretionary manufactured infant sweet or savory snack foods.

Cereal-based products and dishes were the greatest contributors to total energy from discretionary foods with sweet biscuits, and cakes, muffins, scones and cake-type desserts contributing 10.8% and 10.2% of total energy intake from discretionary foods, respectively. Other key contributors to energy intake from discretionary foods included sausages, frankfurters and saveloys (8.3%), vegetable products and dishes where frying was the main cooking technique (8.6%), butter (7.3%), and finally manufactured infant sweet or savory snack foods (9.3%). 

### 3.4. Determinants of Discretionary Food Consumption

In the simple (unadjusted) linear regression analyses, there was a significant association between %TotE from discretionary food and mother’s age (*p* = 0.041), level of education (*p* < 0.001), country of birth (*p* < 0.001), child’s birth order (*p* = 0.003), and age of child when complementary foods were introduced (*p* = 0.001) (Table 4). There was no association with mother’s SEP, the age of the child when she returned to work or the child’s sex.

When all variables with a *p* value ≤ 0.10 were simultaneously entered into the multiple (adjusted) linear regression model, mother’s level of education (*p* = 0.274) and age of the child when they first received complementary foods (*p* = 0.129) were no longer significant (Table 4). The association with maternal age strengthened, as did the association with number of children. Children born to young mothers (<25 years) consumed a higher %TotE from discretionary foods than children born to mothers aged 35 years or older (*p* = 0.008), while singleton children consumed a lower %TotE from discretionary foods than children with two or more siblings (*p* = 0.002). Children of women born in Australia and the United Kingdom consumed a significantly higher %TotE from discretionary foods than children of women born in India, China and other Asian countries (*p* < 0.001).

Removal of 125 participants with implausible energy intakes resulted in similar findings to the primary analysis (Appendix A), with maternal age (*p* = 0.031), number of children (*p* = 0.022) and mother’s country of birth (*p* < 0.001) being the only explanatory factors independently associated with %TotE from discretionary foods.

## 4. Discussion

Despite the Australian Infant Feeding Guidelines recommending that parents should not feed their children discretionary foods when transitioning them to the family diet [9], almost all children (95.7%) consumed some type of discretionary food on one or more days of recording. To our knowledge this is the first study to use the ABS discretionary flag list [29] to estimate in children under 2 years of age the contribution of discretionary foods to total energy intake. Earlier studies of this age group have defined discretionary foods as either ‘extras’ or non-core foods depending on the terminology and classification system used in the ADGs of the day. This may have resulted in some minor differences, although the majority of foods included in these definitions are similar. Nevertheless our findings are consistent with earlier Australian studies which reported that 91% of children aged 12 to 16 months [23] and 99% of children aged 16 to 24 months [22] consumed discretionary foods. Similar to other Australian studies [18,19,22,23], sweet and savory biscuits, hot chips/French fries, and processed meats were the discretionary food items most commonly consumed by members of the SMILE cohort. 

Consumption of discretionary foods by Australian children begins early in the weaning period [18,19,20] and the proportion of children consuming discretionary foods increases markedly in the second year of life [19,24,40]. One in every five children in the second Perth Infant Feeding Study had consumed cakes and biscuits by 22 weeks [18], and the proportions of children in the Melbourne InFANT study consuming sweetened beverages and savory and sweet energy-dense snacks increased more than two-fold between 9 and 18 months [24]. In addition, patterns of discretionary food intake track into later life as demonstrated in the InFANT study, when being a consumer of sugar sweetened beverages and sweet energy-dense snacks, or consuming larger amounts of these foods, at 9 months, was predictive of a greater level of consumption at 18 months [24] and 5 years [21]. Tracking of poor dietary patterns from infancy to preschool age have been reported also in international studies [41,42].

The early and escalating consumption of energy dense-nutrient poor foods in early childhood is not unique to Australian children. In the Dutch BeeBOFT study 20.2% of infants were consuming sweet beverages daily and 16.5% were consuming snack food daily at 6 months of age [43]. The results of the US 2016 Feeding Infants and Toddler Study (FITS) [44] indicate that 34% of infants aged 6 to 11.9 months had consumed sweets such as cakes, biscuits, muffins, frozen desserts or sugar sweetened beverages, with this proportion rising to 73% of children aged 12 to 17.9 months. Similarly, 29% of infants in the UK Diet and Nutrition Survey of Infants and Young Children aged 7 to 9 months had consumed sweet biscuits and 16% had consumed savory snacks, rising to 53% and 28% respectively for infants aged 10 to 11 months [45]. 

Discretionary foods are typically energy dense, and the contribution of discretionary foods to total energy intake increases as consumption of these food increases with age, resulting in the displacement of more nutrient dense foods and beverages [27,46,47]. The average energy intake of the SMILE cohort was similar to that reported for another cohort of Australian children of similarly aged children (12 to 16 months) (4040 kJ vs 4194 kJ), as was the proportion of total energy derived from the consumption of discretionary foods (11.2% vs 9%) [23]. Whereas, Webb et al. [22] reported that 26.5% of total energy consumed by an older cohort of children aged 16-24 months was derived from energy dense-nutrient poor foods. By the time Australian children are aged 2 to 3 years these foods contribute approximately 30% of total energy intake [26], representing a rapid threefold increase from that reported in this study. The proportion of energy in the Australian diet derived from discretionary foods rises steadily throughout childhood, peaking at 41% for children aged 14 to 16 years, thereafter declining to around 35% in adulthood [26].

Discretionary foods are highly palatable, typically being high in either sugar or salt. In this study, just under one quarter of the sodium and almost one half of the free sugars consumed by children were derived from discretionary foods. This is problematic as infants have an innate predisposition for sweet and salty tastes [48]. Adding sugar and salt to foods can drive consumption [3,49] and preferences for high salt/ high sugar foods can be promoted by early and frequent exposure to these foods [48]. There is evidence that excessive intakes early in life of foods high in salt and refined sugars can contribute to a number of conditions in adulthood, such as obesity [11], metabolic syndrome [50], and cardiovascular disease [14,16]. 

Findings of this study are consistent with other studies which have investigated the determinants of dietary behaviors of infants and toddlers and have found that children of younger mothers had less healthy diets than those of older mothers [41,43,51,52,53,54,55,56]. We did not however, find an independent association with discretionary food intake and maternal education reported in other studies [41,51,53,54,55,56]. Nor did we find an independent association between age of introduction of complementary foods and intake of discretionary foods reported in other studies [57]. Both level of maternal education and age of introduction of complementary foods were significantly associated with energy intake from discretionary food in the simple model but not in the adjusted model.

Our finding that first born children consumed less of their total energy from discretionary foods than did children with two or more older siblings is consistent with other Australian [18,58] and international [41,54,55,56] studies which have reported poorer diet quality and dietary patterns among higher birth order children. It has been postulated that older children in the family who are exposed to these foods outside of the home and through advertising may pester their parents for these foods [18], thus increasing the availability of these foods in the household. Additionally, mothers with larger families may be more time poor and therefore rely more on convenience foods which are typically energy-dense and nutrient poor [56]. 

An interesting finding of this study not often investigated in other Australian studies is that children of mothers born in Australia and the UK derived more of their total energy from discretionary foods than did children of women born in Asian countries. Koh et al reported that Australian and British mothers were more likely to introduce hot chips/ French fries early in the weaning period than women born in other countries [18]. Similarly, in an earlier analysis of the SMILE data we reported that at 12 to 14 months of age, sons of women born in Australia were more likely than sons of women born in other countries to exceed the WHO recommendation that less than 5% of total energy should be derived from free sugars [59]; but this association was not significant for daughters.

The Australian IFG are written for health care professionals to help them provide evidenced-based feeding advice when they interact with caregivers. The primary purpose of the guidelines is to promote and support breastfeeding and the emphasis of information regarding the transitioning of children from a milk-based diet to the family diet is on the appropriate age at which to introduce complementary foods and texture progression. There is relatively little information on the specific types and amounts of foods that should be eaten by children. The shortfall in feeding guidance for parents of young children is not limited to the Australian IFG and a recent analysis of a number of national and international feeding guidelines including WHO, European Network for Public Health Nutrition, US and two European national guidelines (UK and France) revealed that recommendations for acceptable consumption levels of sweet and salty food were rarely included [4]. The authors called for guidelines to provide more practical tips for parents, a sentiment echoed by Dwyer [47] who on reviewing the findings of the 2016 wave of the US FITS study suggested that stronger recommendations are needed so that parents understand the specific foods children should and should not be eating.

The infant and toddler food market is a rapidly expanding market and has grown considerably in the last 5 years in terms of both product range and revenue [60]. In this study, one third of children were consuming manufactured discretionary foods marketed as infant or toddler snacks and finger foods which contributed just over 9% of the energy from discretionary foods. In the absence of parent-focused, practical infant and early childhood feeding guidelines, parents look for information from other sources and are susceptible to the advertising of infant food companies who overtly or covertly promote their products to parents as heathy choices for children. This is not necessarily the case as in some instances the free sugars content might be higher than some confectionary. In recent years, advocacy groups in Australia [61,62] and the UK [63] have lodged complaints with the relevant regulatory agencies against Heinz regarding the company’s misleading advertising of two varieties of infant and toddler snacks with excessively high free sugars content, resulting in the discontinuation of these products. 

Products eaten by SMILE children included a variety of sweet and savory snacks such as yoghurt coated mini rice cakes, yoghurt buttons, Little Bellies Animal Biscuits (described as “delightful wholegrain animal-shaped biscuits, to make grumbly tummies happy”), Heinz Little Kids^TM^ range which includes mini tomato corn cakes and yogurt and fruit flavored cereal bars (described as being the “right portion size for little tummies” and “the right size and shape for little hands”), and Healtheries Kids Care^TM^ Potato Stix (described as “delicious crunchy snacks made from potato and rice – perfect for healthy school lunches). Front of pack and website messages such as those used to promote these products misleadingly imply that these foods form a normal part of a child’s diet and development. These products, while for the most part considerably lower in sugar, salt and fat than equivalent adult products, are likely to displace more nutritious fruit and cereal-based snacks. The marketing language used to promote these foods suggest that these snacks, and snacking *per se,* are an important part of a healthy diet and they may act as ‘gateway’ snacks leading to the habitual consumption in childhood and later life of ‘adult’ versions which are high in fat, and added sugars and salt. 

The consumption of discretionary foods by this cohort exceeded recommendations [34]. It has been proposed that rather than focusing on discretionary foods *per se,* that focusing interventions on a targeted group of discretionary choices may deliver the best impact on population nutrient intakes, diet quality and health [27]. An analysis of the 2011-2012 AHS, employing the ABS discretionary flag method, revealed that for children aged 2 to 17 years, cakes, muffins, and slices; sweet biscuits; potato crisps and similar snacks; and, processed meats and sugar-sweetened drinks were relatively commonly consumed, and that cereal-based takeaway foods; cakes, muffins and slices; meat pies and other savory pastries; and, processed meats were top contributors to energy, saturated fat, and sodium across most age groups [27]. These foods were amongst the most commonly consumed discretionary foods in the SMILE study and other Australian studies of toddlers [18,19,22,23]. Therefore these specific foods should be targeted in future iterations of the Australian IFG, which in turn should be supported with materials written specifically for parents providing practical tips for reducing the portion sizes and the frequency of consumption of these foods [27].

The use of 3 non-consecutive days of detailed dietary data is a major strength of this study. A further strength is that discretionary foods were categorized using the discretionary flag list developed and employed by the ABS in the analysis of the AHS, allowing our data to be compared directly with results from the most recent AHS [27]. This overcomes a limitation of earlier studies which have not used a standardized definition of discretionary or ‘extra’ foods [22]. A limitation of this study was that complete dietary data at 12 months of age was available for less than half of the original sample. Nevertheless, the deliberate oversampling of women from more socially-disadvantaged areas resulted in a demographically diverse analysis population that was generally representative of the population from which it was drawn [39]. A further limitation of this study is that information on children’s diets was provided by their mothers and may be subject to social desirability bias resulting in the omission and under-reporting of certain discretionary foods, particularly those recognized by parents as ‘junk’ foods. Thus the level of consumption of discretionary foods reported here may be an underestimation of what is typically consumed by this age group.

## 5. Conclusions

Consumption of discretionary foods begins early in life, and in this cohort of children aged 12 to 14 months more than 10% of energy intakes came from discretionary foods that are high in saturated fat, added sugars and/or sodium. Parents, in particular young mothers and those with larger families, need practical guidance on how much of, and how often, these food should be eaten by their children.

## Figures and Tables

**Table 1 ijerph-17-00080-t001:** Maternal and child sociodemographic characteristics of SMILE participants (*n* = 828).

Characteristics	*n* (%)
**Maternal Characteristics**	
Age at time of delivery (years)	
<25	73 (8.8)
25–34	574 (69.3)
≥35	179 (21.6)
Level of education	
High school/vocational	356 (43.0)
Some university and above	468 (56.5)
Socio-economic position ^(a)^	
Deciles 1–2 (most disadvantaged)	120 (14.5)
Deciles 3–4	173 (20.9)
Deciles 5–6	174 (21.0)
Deciles 7–8	160 (19.3)
Deciles 9–10 (most advantaged)	195 (23.6)
Country of birth	
Australia and New Zealand	610 (73.7)
United Kingdom/ Ireland	31 (3.7)
India	50 (6.0)
China	37 (4.5)
Asia-Other	52 (6.3)
Other	43 (5.2)
Age of child when mother returned to work (months)	
≤6	169 (20.4)
>6–12 months	266 (32.1)
Not working at 12 months	376 (45.4)
Number of children	
1	389 (47.0)
2	291 (35.1)
≥3	121 (14.6)
Pre-pregnancy BMI ^(b)^ (kg/m^2^)	
<25	477 (57.6)
25–24.99	167 (20.2)
≥30	140 (16.9)
**Child Characteristics**	
Sex	
Male	452 (54.6)
Female	376 (45.4)
Birth weight (g)	
<2500	47 (5.8)
2500–4000	684 (83.8)
>4000	86 (10.4)
Age complementary foods introduced (weeks)	
<17	202 (24.4)
17–25	545 (65.8)
≥26	75 (9.1)

^(a)^ IRSAD, Index of Relative Socio-Economic Advantage and Disadvantage, where decile 1 = most disadvantaged and decile 10 = most advantaged. ^(b)^ BMI Body Mass Index.

**Table 2 ijerph-17-00080-t002:** Average energy, total fat, saturated fat, sodium and free sugars from all foods and discretionary foods.

Nutrient	Mean (SD)	Median	Range	25th Percentile	75th Percentile
Energy (kJ) all foods	4040 (954.7)	3987	1636–8599	3360	4645
Energy (kJ) discretionary	474 (445.2)	363	0–2861	151	644
% energy from discretionary	11.2 (9.3)	9.0	0–50.9	4.0	16.4
Total fat (g) all foods	37.7 (11.2)	36.6	11.3–91.9	29.8	44.6
Total fat (g) discretionary	5.7 (5.7)	3.9	0.0–50.0	1.5	8.2
% total fat from discretionary	14.0 (12.1)	10.8	0.0–72.8	4.4	21.5
Saturated fat (g) all foods	17.5 (6.1)	16.8	3.6–45.5	13.1	21.1
Saturated fat (g) discretionary	2.6 (2.8)	1.8	0.0–32.5	0.6	3.8
% saturated fat from discretionary	14.2 (12.6)	11.2	0.0–79.3	4.2	21.0
Sodium all foods (mg)	788.6 (354.4)	736.7	114.3–2128.9	513.0	1011.5
Sodium (mg) discretionary	200.9 (193.6)	142.2	0.0–1134.9	54.5	301.1
% sodium from discretionary	22.6 (16.1)	21.0	0.0–75.9	9.1	34.0
Free(^a)^ sugars (g) all foods	8.8 (9.3)	6.7	0.0–152.7	3.0	12.1
Free sugars (g) discretionary	4.5 (7.3)	2.3	0.0–140.8	0.7	5.8
% free sugars from discretionary	46.1 (32.6)	44.4	0.0–100.0	14.6	74.8

^(a)^ Includes added sugars and sugars naturally present in honey, syrups, fruit juices and fruit juice concentrates.

**Table 3 ijerph-17-00080-t003:** Mean per capita contribution of discretionary food subgroups to total energy (*n* = 828).

Food Group	% of Children Consuming	kJ	kcal	% Total E	% Disc. E
Fruit and vegetable juices and drinks ^(a)^	4.2	15.7	4	0.1	1.1
Bread products ^(b)^	8.9	58.4	14	0.5	4.1
Sweet biscuits	36.6	154.2	37	1.3	10.8
Savory biscuits	18.8	54.4	13	0.5	3.8
Cakes, muffins, scones and cake-type desserts	15.2	145.0	35	1.2	10.2
Pastries	15.0	112.7	27	0.9	7.9
Pizza	3.4	27.8	7	0.2	2.0
Butters ^(c)^	41.5	103.2	25	0.9	7.3
Fish and seafood products ^(d)^	8.6	48.5	12	0.4	3.4
Sausages, frankfurters and saveloys	14.4	118.2	28	1.0	8.3
Processed meat	28.3	67.2	16	0.6	4.7
Frozen milk products ^(e)^	8.5	27.9	7	0.2	2.0
Milk-based desserts ^(f)^	2.5	16.5	4	0.1	1.2
Gravies and savory sauces	28.3	26.5	6	0.2	1.9
Fried Potatoes ^(g)^	19.1	96.3	23	0.8	6.8
Dishes where vegetables are the major component ^(h)^	3.6	26.0	6	0.2	1.8
Snack foods ^(i)^	6.6	15.0	4	0.1	1.1
Sugar, honey and syrups	15.7	19.4	5	0.2	1.4
Jam and lemon spreads, chocolate spreads, sauces	12.2	14.6	3	0.1	1.0
Confectionary and cereal/nut/fruit/seed bars	13.5	46.1	11	0.4	3.2
Chocolate and chocolate-based confectionary	6.9	23.9	6	0.2	1.7
Yeast, and yeast vegetable or meat extracts^j (j)^	36.0	14.9	4	0.1	1.0
Infant sweet or savory snack foods	33.2	132.3	32	1.1	9.3

Consumed sources include ^(a)^ 100% juices and less than 100% fruit and vegetable drinks; ^(b)^ English-style muffins, flatbreads and savory and sweet breads; ^(c)^ plain and unsalted butter, flavored butters, ghee. This does not include margarine or dairy blends (butter and vegetable oils); ^(d)^ homemade and takeaway fried or deep-fried battered or crumbed seafood, and fish or shrimp pastes; ^(e)^ ice-cream, frozen yoghurt and other frozen desserts; ^(f)^ chocolate dairy desserts, fromais frais, panna cotta, and rice pudding (homemade or store bought); ^(g)^ homemade or takeaway deep-fried hot chips/fries, wedges, potato gems, and hash browns; ^(h)^ fried vegetable dishes (e.g. fried fritters, onion rings, fried tempura vegetables); ^(i)^ potato, vegetable and corn chips, popcorn with salt and/or butter, deep-fried prawn crackers and pappadums, salted pretzels, and crackers with processed cheese spread (snack pack) ^(j)^ yeast-based spreads (e.g. Vegemite, Promite).

**Table 4 ijerph-17-00080-t004:** Factors associated with percentage total energy (%TotE) from discretionary foods (mean values and 95% confidence interval) of toddlers (*n* = 828).

Variables	Unadjusted Mean %TotE/day	95% CI	*p*	Adjusted Mean %TotE/day	95% CI	*p*
Total sample	11.2	10.6–11.9				
**Maternal characteristics**						
Maternal age at recruitment (years)			0.041			0.008
<25	13.1 ^a^	11.1–15.2		13.3 ^a^	10.7–15.9	
25–34	11.2	10.4–11.9		10.7	9.3–12.0	
≥35	10.0 ^a^	8.7–11.3		9.1 ^a^	7.4–10.7	
Level of education			<0.001			0.274
High school/vocational	12.4	11.5–13.4		11.4	9.7–13.6	
Some university and above	10.1	9.3–10.9		10.6	9.2–12.1	
IRSAD score			0.151			
Deciles 1–2 (most disadvantaged)	12.0	10.4–13.6				
Deciles 3–4	12.0	10.6–13.3				
Deciles 5–6	9.8	8.4–11.1				
Deciles 7–8	11.1	9.7–12.4				
Deciles 9–10 (most advantaged)	10.8	9.5–12.1				
Maternal country of birth			<0.001			<0.001
Australia and New Zealand	12.1 ^abc^	11.4–12.7		13.1 ^ab^	11.9-14.3	
United Kingdom/ Ireland	13.0 ^def^	9.8–16.1		16.8 ^cde^	13.6–20.1	
India	7.1 ^ad^	4.7–9.4		9.5 ^c^	6.8–12.2	
China	4.5 ^be^	1.8–7.3		6.8 ^ad^	3.7–10.0	
Asia Other	6.3 ^cf^	3.9–8.6		8.8 ^be^	6.2–11.4	
Other	8.7	6.2–11.2		11.0	8.1–14.0	
Age of child when mother returned to work			0.104			
≤6 months	12.0	10.6–13.3				
Between 6 and 12 months	10.2	9.1–11.3				
Not returned to work by 12 months	11.4	10.5–12.3				
Number of children			0.003			0.002
1	10.4 ^ab^	9.5–11.3		9.5 ^a^	8.0–10.9	
2	11.4 ^ac^	10.3–12.4		10.8	9.2–12.5	
≥3	13.6 ^bc^	12.0–15.2		12.7 ^a^	10.7–14.8	
Pre-pregnancy BMI (kg/m^2^)			0.123			
<25	10.7	9.8–11.5				
25–29.99	11.1	9.8–12.5				
≥30	12.4	10.9–13.9				
**Child characteristics**						
Sex			0.433			
Male	11.3	10.5–12.2				
Female	10.8	9.9–11.8				
Age complementary foods introduced (weeks)			0.001			0.129
<17	13.1 ^ab^	11.9–14.4		12.1	10.4–13.8	
17–25	10.6 ^a^	9.8–11.3		10.7	9.2–12.1	
≥26	9.8 ^b^	7.8–11.9		10.3	8.0–12.6	

IRSAD, Index of Relative Socio-Economic Advantage and Disadvantage, where decile 1 = most disadvantaged and decile 10 = most advantaged. BMI Body Mass Index. ^a,b,c,d,e^ A shared superscript indicates a significant difference between the groups.

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
