# Peer review of "Sources and Determinants of Discretionary Food Intake in a Cohort of Australian Children Aged 12–14 Months"

_ijerph, 2019, doi:10.3390/ijerph17010080_

Round 1

Reviewer 1 Report

Sources and determinants of discretionary food intake in a cohort of Australian children aged 12-14 months

The study focuses on an important research question. In general, it was well designed and adequately conducted and written. However, some aspects should be clarified and/or improved, as indicated below.

Methods:

- It would be very interesting to know the results regarding the food processing method (NOVA). But, I understand that following the Australian guidelines it was not possible for this study. I suggest the authors in the future work with NOVA classification displaying the amount of energy consumed by ultra-processed foods. 

- Please, provide more details about the recruitment and follow-up. Regarding the factors associated included in the analyzes, how were they collected?

- Provide the sample size calculation

- Line 92: Provide reference for “five-steps multiple pass method”

Results:

- Table 1: One suggestion – include more information to describe the children (e.g., birth characteristics, nutritional assessment)

- Table 2: I suggest showing the results also in kcal to better understanding.

- Table 3: Please clarify what was considered for the group “Fruit and vegetable juices and drinks”.

Discussion:

- The authors need to include the losses during follow-up in the limitation section.

Author Response

Thank you for your positive response and constructive feedback. 

Methods:

- It would be very interesting to know the results regarding the food processing method (NOVA). But, I understand that following the Australian guidelines it was not possible for this study. I suggest the authors in the future work with NOVA classification displaying the amount of energy consumed by ultra-processed foods.

Response: We agree that it would be interesting to investigate the level of processing of the diets of this cohort of children and will certainly investigate the possibility of conducting this analysis using the NOVA classification system. 

- Please, provide more details about the recruitment and follow-up. Regarding the factors associated included in the analyzes, how were they collected?

Response: Further details on follow-up and data collection are provided (lines 97-102)

- Provide the sample size calculation

We have chosen not to provide a sample size calculation for this secondary analysis as the sample size calculation for the SMILE study (below) was based on the primary outcome variable i.e. mean decayed, missing and filled surfaces (dmfs) and therefore was not applicable to the analysis in this paper. Nevertheless, the size of this cohort with complete dietary data (n=828) is considerably larger than other Australian cohorts Nourish (n=551) (Byrne et al. 2014) and InFANT (n=467) (Spence et al. 2018) and large enough to detect significant differences between groups.

“Sample size was calculated to detect a rate ratio of 0.2 between slopes for explanatory variables in multivariable regression models for caries experience (mean dmfs) at age two years with an alpha level of 0.05 (two-tail) and statistical power of 90%. The calculated sample size required at age two years is 1,398 children. A minimum two-year retention rate of 80% was used in the calculation. It resulted in a targeted sample size at birth of 1,677 children (rounded up to 1,700). This recruitment target is highly achievable given the population pool of 16,000 children. The expected two-year retention rate of 80% is considered conservative based on our experience. Using this sample size, statistical power expected to achieve other aims was calculated at over 90% after taking into account possible interaction among variables.” (Do et al. 2014)

- Line 92: Provide reference for “five-steps multiple pass method”

Response: reference added – line 109

Results:

- Table 1: One suggestion – include more information to describe the children (e.g., birth characteristics, nutritional assessment)

Response: As the SMILE study was primarily an ORAL health study only birth weight was recorded at baseline and this has been added to table 1.  No nutritional assessment was undertaken until the time of the dental examination which was conducted between the child’s 2nd and 3rd birthday. At this time weight and height were measured using standardised procedures but they are not reported here as they are not relevant to this analysis of children when they were aged 12-14 months.

- Table 2: I suggest showing the results also in kcal to better understanding.

Response: The kj content were divided  by 4.2 to generate mean kcal and added to table 2.

- Table 3: Please clarify what was considered for the group “Fruit and vegetable juices and drinks”.

Response: A footnote has been added to the table. This category included 100% fruit and vegetables juices and drinks which included less than 100% fruit and vegetable juice (often described by parents as juice).

Discussion:

- The authors need to include the losses during follow-up in the limitation section.

Response: This limitation has been added to the discussion (lines 436-439).

Reviewer 2 Report

The manuscript is generally well written, describing sources and determinants of discretionary food intake in a cohort of Austalian children aged 12-14 months.

1) The results are in line with previous studies in Australia with similar objectives and I miss at least an attempt to hightlight the novelty of this study/analysis in the introduction section.

2) Please include some information about the results on determinants of discretionary food intake in the abstract (now only included in the conclusion part).

3) More detailed information on how discretionary foods are defined by ADGs (lines 49-50) and this compared with the approach used in the present analysis (mentioned in the discussion, line 368). More detailed information is needed in the method section about the discretionary flag list in the method section (lines 107-112), with examples on how much sugar, salt or fat a product must contain to be listed. More information is also needed on how the infant foods were treated. A supplement table of the 187 infant food products would be helpful, including the criteria why they are considered discretionary foods. This might be useful for the authorities when updating the recommendations for parents of infants (as you propose in the discussion) and for future studies for comparison.

4) You apply two different methods when assessing dietary intake. Could you comment on why you chose this method and not simply 3day food records? Have the portion size images and household measures included in the food record booklet been validated? Did you obtain similar portion sizes from using the 24hour recalls and the 3d food records?

5) According to information in the result section disadvantaged women were over-recruited (line 171). Please add some information in the method section on how this was done (where it says that mothers with sufficient English competency and those who did not intend to move were invited).

6) In the discussion you compare your results to results from other Australian cohorts. Did they apply the same methodology as you did (the ADGs list).

7) Information on the average intake of total energy, free sugars, salt, fat and saturated fat should be reported, compared to current recommendations and included in the discussion. How bad is the total diet in general? Do you think that all these food items should be omitted from the infant diet? Some children (with low appetite) might need extra fat to increase their energy intake and during sick days parents might use whatever necessary to get energy into their children. Do you have any information on general health, growth or appetite of the children in this cohort?

Author Response

Reviewer 2

Thank you for your constructive feedback

The results are in line with previous studies in Australia with similar objectives and I miss at least an attempt to highlight the novelty of this study/analysis in the introduction section.

Response: Children under 2 years of age are not included in the National Health survey, therefore we know relatively little about the diets of this age group as they transition from a milk-based diet to the family diet.  While other researchers have reported on the intake of non-core or ‘extra’ foods in this age group, none have used the methodology employed by the Australian Health Survey to define and flag discretionary foods, thereby allowing direct comparison with the AHS findings related to older children in the AHS. The following sentence has been added to the introduction to highlight the novelty of the study (lines 78-80) .

This is the first study of Australian children under the age of 2 years of age to use the methodology employed in the AHS to define and flag discretionary foods, allowing direct comparison with the intakes of older Australian children.

 Please include some information about the results on determinants of discretionary food intake in

Response: This information has been added to the abstract.

More detailed information on how discretionary foods are defined by ADGs (lines 49-50) and this compared with the approach used in the present analysis (mentioned in the discussion, line 368).

The general definition of discretionary foods, as defined by the ADGs is provided in lines 51-55. A text box with the principles used to classify discretionary foods is now included in the manuscript and provides more detail. We employed this method in this study as clarified in lines 128-130. To date the only other Australian study to clearly report using this method to classify discretionary foods is a secondary analysis of the NHS data for children 2-17 years (Johnson et al 2017). We trust that with the changes made it is now clear to the reader that this was the method used in the current study.

More detailed information is needed in the method section about the discretionary flag list in the method section (lines 107-112), with examples on how much sugar, salt or fat a product must contain to be listed. More information is also needed on how the infant foods were treated.

The text box with the principles and cut-off criteria used by the Australia Bureau of Statistic to classify discretionary food has been included in the manuscript. This same method was used to assess and classify the 187 infant foods not in eh AUSNUT database, as already stated (lines 131-133).

“The 187 infant foods not listed in the AUSNUT 2011-13 food composition database or the Australian Bureau of Statistics (ABS) discretionary flag list, were individually assessed and categorized as discretionary or non-discretionary using the same criteria as outlined in the AHS User Guide.”

A supplement table of the 187 infant food products would be helpful, including the criteria why they are considered discretionary foods. This might be useful for the authorities when updating the recommendations for parents of infants (as you propose in the discussion) and for future studies for comparison.

Response

A supplementary table of the 187 infant food products is included flagging those identified as being discretionary foods (1 = yes).

You apply two different methods when assessing dietary intake. Could you comment on why you chose this method and not simply 3day food records?

Response: Assessment of dietary intake of very young children is problematic due to high levels of plate waste, frequent consumption of small amounts of food, and the need for proxy reporting completed by a parent or caregiver.   Food Frequency questionnaires are the less burdensome but typically overestimate energy and nutrient intakes.  The food record is associated with greater accuracy however carries a high participant burden.  On the other hand the 24 hr recall relies on trained interviewers and repeat 24 hr recalls are costly to administer. The combined 24 hr recall and 2 day food record has been used in other studies (Nourish) and has the advantage of reducing both participant and investigator burden.  The multiple pass 24 hr recall provides participants with an indication of the level of detail required when completing the 2  day FR  and the  more detailed information on food type (e.g. white versus wholegrain bread) collected in the 24 hour recall can be used to impute missing information from the food record if necessary. The reason for using this method is provided in the opening sentence of line 104.

Have the portion size images and household measures included in the food record booklet been validated? Did you obtain similar portion sizes from using the 24hour recalls and the 3d food records?

Response: The portion size images and household measures were used previously in the 2007 National Children’s Nutrition and Physical Activity Survey and the 2011-12 National Health Survey. This information has been added to the methodology (Iines 110-115). While we have compared the relative validity for key nutrients of the 24 hr recall against the combined three days of dietary data we have not compared the portion size estimates from using the 24 hr recall and the 2 day food record.

Beaton, E., J. Wright, G. Devenish, L. Do and J. Scott (2018). "Relative validity of a 24-h Recall in assessing intake of key nutrients in a cohort of Australian toddlers." Nutrients 10(1).

According to information in the result section disadvantaged women were over-recruited (line 171). Please add some information in the method section on how this was done (where it says that mothers with sufficient English competency and those who did not intend to move were invited).

Response Additional information on the oversampling procedure has been added to the methods section (lines 93-96).

In the discussion you compare your results to results from other Australian cohorts. Did they apply the same methodology as you did (the ADGs list).

We have clarified that to our knowledge that this is the first study of children under 2 years to apply the ABS discretionary flag list. Earlier studies have used terminology and classification systems relevant to the ADGs of the day, which may have resulted in minor variation.

Information on the average intake of total energy, free sugars, salt, fat and saturated fat should be reported, compared to current recommendations and included in the discussion. How bad is the total diet in general?

The focus of this paper was limited to discretionary food intake. The purpose of this analysis was not to report on the overall quality of the diet of this cohort of children, as more detailed analysis of dietary patterns including a dietary quality index has been reported in a previous paper. Bell, L. K., C. Schammer, G. Devenish, D. Ha, M. W. Thomson, J. A. Spencer, L. G. Do, J. A. Scott and R. K. Golley (2019). "Dietary Patterns and Risk of Obesity and Early Childhood Caries in Australian Toddlers: Findings from an Australian Cohort Study." Nutrients 11(11): 2828.

Do you think that all these food items should be omitted from the infant diet? Some children (with low appetite) might need extra fat to increase their energy intake and during sick days parents might use whatever necessary to get energy into their children.

Response We believe that it is clear from the introduction (lines 55-58) and the tone of the discussion that our views are aligned with the Australian Dietary Guidelines, which state that discretionary foods are not necessary for a healthy diet and that their intake should be limited.  As we did not ask about a child’s health on the day that dietary intake was recorded we cannot say with certainty whether some parents were using these foods as a means of getting energy into sick children.  It is unlikely that large number of mothers would have been employing this tactic, and the consistency of findings with earlier studies suggests good external validity.

Do you have any information on general health, growth or appetite of the children in this cohort?

Response: Weight and height were measured at the time of the dental examination when children were aged between 2 and 3 years, and is reported in the analysis of the earlier paper by Bell et al.2019.